# Molecular, Evolutionary, and Structural Analysis of the Terminal Protein Domain of Hepatitis B Virus Polymerase, a Potential Drug Target

**DOI:** 10.3390/v12050570

**Published:** 2020-05-22

**Authors:** Timothy S. Buhlig, Anastasia F. Bowersox, Daniel L. Braun, Desiree N. Owsley, Kortney D. James, Alfredo J. Aranda, Connor D. Kendrick, Nicole A. Skalka, Daniel N. Clark

**Affiliations:** 1Microbiology Department, Weber State University, 1415 Edvalson St., Ogden, UT 84408, USA; timothybuhlig@mail.weber.edu (T.S.B.); desireeowsley@mail.weber.edu (D.N.O.); kortneyjames@mail.weber.edu (K.D.J.); alfredoaranda@mail.weber.edu (A.J.A.); connorkendrick@mail.weber.edu (C.D.K.); nicoleskalka@mail.weber.edu (N.A.S.); 2Biology Department, Lebanon Valley College, 101 N. College Ave., Annville, PA 17003, USA; afb001@lvc.edu (A.F.B.); db008@lvc.edu (D.L.B.)

**Keywords:** hepatitis B virus, terminal protein, protein priming

## Abstract

Approximately 250 million people are living with chronic hepatitis B virus (HBV) infections, which claim nearly a million lives annually. The target of all current HBV drug therapies (except interferon) is the viral polymerase; specifically, the reverse transcriptase domain. Although no high-resolution structure exists for the HBV polymerase, several recent advances have helped to map its functions to specific domains. The terminal protein (TP) domain, unique to hepadnaviruses such as HBV, has been implicated in the binding and packaging of the viral RNA, as well as the initial priming of and downstream synthesis of viral DNA—all of which make the TP domain an attractive novel drug target. This review encompasses three types of analysis: sequence conservation analysis, secondary structure prediction, and the results from mutational studies. It is concluded that the TP domain of HBV polymerase is comprised of seven subdomains (three unstructured loops and four helical regions) and that all three loop subdomains and Helix 5 are the major determinants of HBV function within the TP domain. Further studies, such as modeling inhibitors of these critical TP subdomains, will advance the TP domain of HBV polymerase as a therapeutic drug target in the progression towards a cure.

## 1. Introduction

Hepatitis B virus (HBV) infects the liver and is commonly spread through sexual, blood, and vertical contact. It is a leading cause of viral hepatitis, liver cirrhosis, and hepatocellular carcinoma. Although several hundred million people in the world are infected by HBV, an effective vaccine exists, as well as non-curative treatment [1]. Current antiviral drugs, except immunomodulatory drugs such as pegylated interferon α and thymalfasin, are all directed against the HBV polymerase (Pol) [2,3,4], though therapeutics for novel targets are in development [5]. Although there is no small animal model capable of infection with human HBV, informative studies have been performed using related HBV species that infect avian and mammalian hosts, such as duck HBV (DHBV) and woodchuck HBV [6]. With the discovery of the receptor for HBV (human sodium taurocholate co-transporting polypeptide receptor (hNTCP or *SLC10A1*)) [7], the ability to create transgenics in small animals and cell culture systems is also leading to advances in model systems [8].

HBV contains a small circular DNA genome, reverse transcribed by the HBV Pol from viral RNA [9]. The HBV Pol protein is a major focus of basic and translational research; indeed, it is the only current target of any HBV-specific antiviral and the only region commonly sequenced during treatment escape [10]. Although the catalytic activity of HBV Pol occurs within the RT and RNase H domains (Figure 1), the TP domain also shows considerable functional utility. The TP domain is unique to *Hepadnaviridae*, and therefore, any drug cross-reactivity would likely be low.

Downstream of the TP domain are the spacer, reverse transcriptase (RT), and RNase H domains (Figure 1). HBV Pol is similar in sequence and structure to the polymerase found in the human immunodeficiency virus (HIV,) which also contains RT and RNase H domains. Indeed, emtricitabine, lamivudine, and tenofovir are able to inhibit both HBV and HIV [9]. Much more is known of the HIV Pol, including several high-resolution structures. Due to their similarity, models of HBV Pol borrow this knowledge about the HIV Pol [11,12]. Despite similarities, definitive descriptions of the three-dimensional structure of HBV Pol do not yet exist. Additionally, HBV’s TP and spacer domains are unique to the Hepadnaviridae family. Currently, their structures can only be predicted.

TP proteins also exist in other microbes; however, they give clues to the function—not necessarily the structure—of HBV Pol. For example, bacteriophage, members of the Adenoviridae family, and *Streptomyces* bacteria are groups of microbes that encode a TP protein for priming DNA synthesis [13,14,15]. Typical among these TP proteins is the use of a tyrosine, serine, or threonine for initiating priming [16,17]. Beyond these priming residues, little amino acid homology can be identified among TP proteins (Figure 1). One commonality among TP proteins is the presence of a disordered priming loop in their protein structure, whose flexibility allows access to the active site of DNA synthesis proteins. However, other than in Hepadnaviridae, all TP proteins exist separately from the catalytically active polymerase protein. The polymerase in Hepadnaviruses synthesizes both DNA strands while still attached to the DNA [18].

Determining the three-dimensional structure of the TP domain has thus far proved impossible. Reasons include the difficulty of purifying large amounts or truncated portions of HBV Pol for crystallography. Additionally, the structure is disordered in several places, and the protein may exist in several conformations [19]. The conformation of HBV Pol varies during the several stages of DNA synthesis and is maintained by both host chaperone proteins and its association with an RNA secondary structure element called epsilon (ε RNA). An initial conformation is provided when the host chaperone proteins Hsc70, Hsp40, Hsp90, and Hop bind to HBV Pol [20,21]. Only this chaperone-associated Pol protein is capable of binding ε RNA. The subsequent binding of ε RNA induces another conformational change in HBV Pol before DNA synthesis, allowing the delivery of the Y63 priming residue to the active site in the RT domain [22]. The conformation changes after priming, as evidenced by protein cleavage assays in DHBV and the finding that continued synthesis of viral DNA does not continue along the 5′ ε RNA but is instead templated by the 3′-end of the pgRNA after template switching [23,24]. These facts suggest that the structure of the polymerase may not be amenable to crystallography. Other means of determining structure have been performed, including epitope mapping with the analysis of antibody binding sites [25,26]. Technologies such as high-resolution mass spectrometry, nuclear magnetic resonance spectroscopy, and cryogenic electron microscopy may prove useful in determining a structure.

A functional cure for HBV would likely require combinations of drugs that target multiple non-redundant targets, perhaps including the TP domain of HBV Pol. Chronic HBV infections are currently treated with pegylated interferon-α, which increases immune activity, and/or with nucleoside analogs that block the RT domain from replicating viral DNA. Both of these main classes of drugs achieve hepatitis B surface antigen (HBsAg) loss only rarely; interferon is poorly tolerated, and nucleoside analogs are administered for life to achieve a reduction in viral load [2,3,4].

Several drugs are in clinical trials, and drug development against HBV is an exciting realm of possibility. Techniques for drug development include 3D in silico modeling that attempts to dock libraries of chemicals to viral proteins. Candidates may be chosen for cell-based or cell-free assays [27,28,29,30]. Cell based assays are more physiologically relevant, have the ability to test toxicity, and usually involve cells permissive to HBV (or transfected HBV DNA) such as primary human hepatocytes, HepG2, Huh7, HepaRG, or others [8,31]. Cell-free assays, on the other hand, require purified cellular and/or viral components but provide a higher throughput capacity, such as using purified HBV Pol to measure elongation activity in vitro [32]. Other specialized techniques such as split luciferase or other two-factor interaction tests may reveal molecular partnerships [33].

Functionally essential and chemically unique, the TP domain’s potential as a therapeutic target is high. Therefore, an analysis of current research was performed to map the TP domain of HBV Pol. These studies evaluate the role of specific amino acid residues in four of the primary functions of the polymerase: RNA binding, protein priming, RNA packaging, and DNA synthesis. In addition, methods for evaluating these four main functions are described. The functional mapping of specific regions within HBV Pol’s TP domain is discussed, namely, conservation analysis, secondary structure prediction, and targeted mutational studies. With no solved structure for HBV Pol, this analysis provides a valuable map of the TP domain. However, this mapping may be superseded once a suitable high-resolution three-dimensional structure is solved alongside its molecular interactions.

## 2. Four Main Functions of HBV Pol Determined by the TP Domain

Functional assays in use in laboratory settings are able to evaluate the critical enzymatic effects and interactions of HBV Pol. For example, the TP domain contains the tyrosine Y63, whose hydroxyl group provides the substrate for initiating DNA synthesis, called protein priming [34]. Priming occurs with the TP domain acting as the primer, and the viral pgRNA as a template. Near the 5′-end of the pgRNA, the ε RNA stem-loop structure is recognized in a sequence-specific manner and bound by HBV Pol, thus acting as the packaging signal. The shape of the RNA molecule provides an unpaired bulge, which is the template for DNA synthesis [35,36]. Thus, all four main functions of HBV Pol, discussed in detail below, are dependent on the TP domain (Figure 2).

Although DNA synthesis and RNase activity are provided directly by the RT and RNase H domains, respectively, the TP domain is critical for several essential steps, making it a potential drug target. Figure 2 details these steps within an infected cell, as well as the laboratory assays to measure interactions and activity levels. These assays are most useful in testing mutated versions of HBV Pol to pinpoint critical regions or amino acid residues.

### 2.1. Evaluating RNA Binding of HBV Pol

The earliest step in creating the viral genome begins when the pgRNA becomes associated with HBV Pol. Several in vivo and in vitro assays have been performed in both duck and human HBV to measure RNA binding. Binding assays have mostly been studied in DHBV systems, where HBV Pol is expressed in an in vitro translation reaction in the presence of radiolabeled ε RNA, or a filter binding assay that allows soluble ε RNA to bind to immobilized HBV Pol protein [25,37,38]. For these studies, the RNA is produced in vitro and can be radiolabeled. The amount of RNA is measured after HBV Pol, and the in vitro-generated RNA is allowed to bind.

To assay the RNA binding of human HBV in cell culture, the HBV Pol protein, as well as an ε RNA construct, can be expressed in human cells [34]. RNA binding likely occurs in the cytoplasm, and during the RNase-free purification of HBV Pol, the ε RNA remains attached. Both in vivo and in vitro studies usually detect bound viral RNA by measuring radiolabeled RNA levels, or RNA is detected by Northern blotting with a probe specific for viral RNA (Figure 2b).

### 2.2. Evaluating Protein Priming of HBV Pol

There are several ways in which the TP interacts with the RT domain after RNA is bound. For the initiation of DNA synthesis in the protein priming step, the TP domain’s amino acids would need to deliver functional groups such as hydroxyls to the RT domain’s catalytic site, allowing them to act as a primer and substrate for reverse transcription. Modeling has predicted that the region surrounding the Y63 priming site is within an unstructured protein loop or may be part of a beta strand [39,40,41,42]. This flexibility may be needed for the priming loop to enter the catalytic cleft in the RT domain. By deleting ~10 amino acid regions of this priming loop, all priming activity was blocked [39]. This may be because these shorter loops were unable to reach into the RT active site.

Several experimental conditions can test these initial steps needed to synthesize the relaxed circular DNA (rcDNA) genome of HBV. Duck HBV Pol has been used extensively in vitro, where an *E. coli* expression system produces a priming-active polymerase [43]. The DHBV system removes non-essential protein regions and is an informative assay of HBV Pol function. However, DHBV Pol priming is not ε RNA-dependent and not template-dependent (and thus represents transferase activity instead of template-based replication). Hence, the duck HBV priming system may not accurately represent in vivo conditions.

Even by reconstituting chaperone proteins and ε RNA, the human HBV polymerase is not active in in vitro priming. These fastidious requirements hampered in vitro studies of human HBV Pol priming until a system reproduced human HBV polymerase activity in ε RNA-dependent protein priming [34]. In this system, HBV Pol is co-expressed in cell culture with ε RNA, and the two associate. HBV Pol and any associated chaperone proteins can then be purified by the immunoprecipitation of HBV Pol. This in vitro assay is performed under RNase-free conditions, allowing the enzymatically active polymerase to use bound ε RNA as a template for initiating DNA synthesis in an in vitro priming reaction. In vivo, the protein priming reaction likely occurs within the nucleocapsid, whereas in vitro, the priming occurs in the absence of capsid proteins. Only the first few nucleotides are synthesized, similar to what is thought to occur before template switching [24]. Once purified, radiolabeled nucleotides are supplied in a buffer containing either magnesium or manganese as the enzyme co-factor [44]. Priming is measured by the autoradiography of the polymerase protein, since priming attaches the radionucleotides to itself (Figure 2b).

On a related note, when manganese is used as the enzyme co-factor, in vitro priming by human HBV Pol occurs in a non-template-directed fashion, called transferase activity. Transferase activity tends to be more processive—instead of synthesizing only a few nucleotides, strands of several hundred nucleotides are made [44]. Currently, there is no evidence of manganese priming or transferase activity in vivo, but this assay may be useful for drug screening.

### 2.3. Evaluating RNA Packaging and DNA Synthesis of HBV Pol

Assays to study hepatitis B virus activity in RNA packaging and DNA synthesis evaluate the nucleic acid content of purified nucleocapsids. These experiments have been performed in both duck and human HBV [45]. In the HBV replication cycle, RNA packaging is the process by which RNA is incorporated into a viral nucleocapsid. Capsid subunits self-associate, encapsidating the pgRNA and HBV Pol (Figure 2a). Packaging occurs due to interactions between HBV capsid and Pol proteins and pgRNA, as well as with host proteins [46]. Nucleocapsids can be assayed for their nucleotide content—RNA or DNA—usually after conducting agarose gel electrophoresis of the intact capsids. A Western blot is also used to measure capsid levels, and the ratio of capsid protein is compared to the nucleotide content [47]. The specificity of this assay lies in the choice of the probe used for blotting—it must detect RNA or DNA, but not both (Figure 2b).

DNA synthesis refers to the elongation of the minus strand. The DNA is synthesized from the pgRNA template inside the maturing virion; RNA degradation occurs concurrently. DNA synthesis also refers to the subsequent step, wherein the second strand is synthesized. HBV is known to contain an incomplete plus strand of DNA, copied from the complete minus strand. These steps are completed before the virus is secreted [48]. To measure DNA synthesis, DNA from intact nucleocapsids (or released from nucleocapsids) is measured by Southern blotting. An alternative method, the endogenous polymerase reaction, allows the completion of the plus strand using radiolabeled nucleotides that are detected by autoradiography [41].

### 2.4. Evaluating Other HBV Pol Functions

The ability to dissect each stage of DNA synthesis, including interactions with template RNA, allows the functional testing of HBV polymerase mutants on a step-by-step basis. Other examples of useful HBV Pol assays include tests of polymerase conformation using conformation-specific antibodies and tests of interactions between the TP and RT domains using *trans*-priming systems [49]. The TP domain is also the site of cleavage when the entire HBV Pol protein is removed from the viral DNA by the ubiquitin-mediated degradation of HBV Pol [50]. The measurement of “protein-free” viral DNA is useful for evaluating the source of covalently closed circular DNA, which resides in the nuclei of infected cells [51].

Fluorescence microscopy can reveal intracellular localization—the TP domain contains mitochondrial localization signals, with the strongest signal in amino acids 141–160, corresponding to Helix 5 and the T3 loop [52]. Additionally, the TP domain contains several epitopes that may be processed and presented to T cells during the adaptive immune response to HBV [53]. Clinical outcomes may result from genetic variation within the TP domain, such as one that may be predictive for fulminant hepatitis infections [54].

However, some functions do not have a suitable assay, including the template switching step that is required after the initial priming. Additionally, since there is no DNA synthesis known to occur until encapsidation [49], the TP may receive a yet-to-be-described activation signal after capsid formation. Other functions that are more directly related to other domains (the RT and RNase H domains) are outside the scope of this review.

## 3. Generating a Structure—Function Map of the TP Domain of HBV Pol

Considering the functional importance of the TP domain and lack of high-resolution structural data, a functional map of structure–function relationships in the TP domain has recently been compiled [39]. This analysis was able to divide the TP domain into subdomains according to their predicted secondary structure. By comparing the critical functions of each subdomain, as found by mutational analyses, three functional loops have been defined. A conservation analysis corroborates the practical importance of these loops among HBV isolates from all known hosts (Figure 3c). Combining conservation analysis, secondary structure prediction, and data from mutational studies gives a map of structure–function relationships to allow for targeted therapeutic design.

### 3.1. Conservation Analysis of TP Domain of HBV Pol

Conservation analysis relies on the premise that conservation occurs only where evolutionarily significant DNA is found. Hepatitis B virus species infect several mammalian [56,57,58,59,60,61], avian [62,63,64,65,66,67], and piscine hosts [68,69]. Using amino acid sequences covering only the TP domain, a phylogenetic tree was generated for known human sequences and several HBV species (Figure 3a). Human HBV is most closely related to the hepatitis B viruses that infect primates, followed by bats, rodents, birds, and finally, fish.

Amino acid sequences were also evaluated phylogenetically for all human isolates with a complete TP domain. Human HBV isolates are usually grouped into genotypes A through G [70,71,72,73,74,75]. Interestingly, when generating the phylogenetic tree using only sequence data for the TP domain (in contrast to using the entire Pol sequence), distinct branching still appeared along classical genotypes (see colored branches in Figure 3b), demonstrating that the TP domain diverges along with genotype groups, a sign of the evolutionary importance of the entire domain. Additionally, specific loop subdomains within the TP domain are functionally important.

When TP domain conservation among known human isolates is graphed along the amino acid sequence (Figure 4a, black graph line), the highest levels of conservation are found in loop regions. The most conserved amino acids among DHBV and HBV isolates (Figure 4a, yellow stars) were also primarily in loop regions. Since conservation is highest in the loops, they may be more evolutionarily important, whereas helices are more divergent. This pattern, where conserved residues are only found in loop regions, is also seen when comparing divergent HBV species. The alignment of mammalian, avian, and piscine sequences revealed that among the 27 amino acids that are 100% conserved in the TP domain, all but three are found in loops (H140, L148, and L151, which are all found in Helix 5) (Figure 3c).

### 3.2. Prediction of Secondary Structure of TP Domain of HBV Pol

Representative amino acid sequences for several HBV species were used to predict secondary structures using SABLE [76], matching previously published secondary structure predictions [39,42]. The sequences have relatively low similarity. For example, between the duck and human HBV TP domains, there is <50% nucleotide sequence identity and 39% protein similarity. Despite this primary structure divergence, the predicted mammalian, avian, and piscine secondary structures have overlapping alpha helices and beta sheets and discernable priming loops, L2 packaging loops, and T3 RNA binding loops (Figure 3c and Figure 4a).

Additionally, a predicted three-dimensional model of the TP domain of human HBV was also produced with QUARK [39,77], which helps to visualize these loops (Figure 4b). Although this prediction does not represent actual three-dimensional structural data, it facilitates the visualization of potential interactions. Loops may be more conserved than helices because HBV Pol is an enzymatic globular protein, not used for structure per se; therefore, the shape and not the sequence has been conserved in the more rigid helical regions. The helices may act as structural anchors, while the loops extend outward and interact functionally with other domains, viral nucleic acids, or host factors.

### 3.3. Mutational Studies of TP Domain of HBV Pol

All known studies of HBV mutants and DHBV mutants (where amino acids are conserved between the duck and human HBV) were considered in an analysis and are represented in Figure 4a. The results summarize the results of four primary functional tests (Figure 2b). The secondary structure boundaries coincide with defective mutants—almost all the defective mutants are found in loops, further supporting the structure–function map.

According to the results of mutational analyses, each loop subdomain (Figure 4a) has been assigned to specific replication steps of the HBV replication cycle (Figure 2a). Amino acids at positions 41 to 81 were defined as the priming loop, not only because this subdomain contains the actual priming site, Y63, but because the known and novel mutants in these regions were defective at the priming step of the viral replication cycle, but not usually at previous steps. The L2 packaging loop spans amino acids 97 to 134; mutants here affect mostly RNA packaging and downstream activity. The T3 loop spans positions 153 to 174 and contains the T3 motif (positions 153 to 160), which was shown by mutational analysis to be critical for RNA binding—the earliest stage in viral DNA synthesis (Figure 4a).

Overall, testing supports the stepwise organization of these four main steps of rcDNA synthesis—RNA binding occurs before packaging and priming, and priming occurs before DNA synthesis. When mutant phenotypes are observed, a defect in an earlier step generally inhibits later steps. With rare exceptions, attributed to assay artifacts [39,78], all mutants that were defective in RNA binding were defective in all downstream activity. Likewise, all priming-defective mutants were also defective in downstream DNA synthesis.

The compiled mutations all represent substitution mutants (Figure 4a). However, internal and *N*-terminal truncation mutants were also constructed to determine the minimal necessary regions for HBV Pol function. Only the first 13 amino acids are dispensable for protein priming activity, and the first 66 amino acids are dispensable for RNA binding activity [35,39,79].

#### 3.3.1. Mutations in the *N*-Terminal Helices

There are likely three small helices within the first subdomain of HBV Pol, spanning the first 40 amino acids (Figure 4a). Mutations made on the *N*-terminal helices subdomain have been created mainly on charged residues, such as the basic amino acids R9/R10 and R35/R36, and the acidic amino acids at positions D15/D16, E22/E23, D30/E31, and E39/D40. Acidic residues are potentially crucial in coordinating binding to the negatively charged phosphate backbone on nucleic acid strands. When mutated to alanines by pairs, these mutations induced no defects in RNA packaging or DNA synthesis [80].

The effect of a mutation on amino acid E31 was evaluated for its impact on RNA binding, protein priming, RNA packaging, and DNA synthesis. E31 is part of a trio of acidic residues—D30, E31, and D32. It is 100% conserved in DHBV, although 41% of human HBV isolates contain an alanine at position 31. The glutamic acid was changed to lysine (E31K), reversing the charge. Despite the reversed charge, this mutant showed only mild defects in priming [39].

Overall, no mutations in the *N*-terminal helix subdomain have caused any phenotypic defects. One potential amino acid, L33, is 100% conserved in avian and mammalian HBV but has not been tested by mutational analysis (Figure 3c). Although no point mutants affect function in the *N*-terminal region, larger truncations do affect protein priming and, presumably, downstream DNA synthesis. When using in vitro assays, truncating the first 20 amino acids caused a loss of priming activity [79], and the removal of even the first 13 amino acids partially reduced the levels of protein priming [39]. Perhaps the reason the smaller substitutions showed no defect is that a large number of acidic residues work in a coordinated fashion, which resists small changes. The *N*-terminal region as a whole is necessary for function, but the exact amino acid positions have not yet been identified. It may be part of a folding core, supporting the flexible loop regions.

#### 3.3.2. Mutations in the Priming Loop

The subdomain from positions 41 to 81 was dubbed the priming loop because most defective mutants in this loop were able to perform upstream activities—binding and packaging RNA—but they were not capable of protein priming or downstream DNA synthesis.

The priming loop is also so named because it contains the Y63 priming site, where a free hydroxyl group attaches to the 5′ phosphate group on the incoming nucleotide during the first step of DNA synthesis [81]. The Y63 priming site is within a beta sheet, which is predicted to exist in the priming loop of mammalian, avian, and piscine HBV (Figure 3c and Figure 4a). The priming loop likely folds together at this locus, allowing it and the ε RNA to fit into the active site of the RT. Both the ε RNA and the priming loop must enter into the catalytic cleft of the RT domain for synthesis to begin [82].

The H54 and K55 sites were predicted to be in a beta strand directly opposite the Y63 priming residue. Interestingly, when these two sites were mutated, H54A showed no defect, while the K55E mutant showed a moderate priming defect (55% of wild-type) [39]. If in a beta sheet, their side chains would face opposite directions; therefore, the different phenotypes support the beta sheet prediction. H54 is 100% conserved among all avian and mammalian HBV isolates, and K55 is conserved among many mammalian HBV isolates.

Near the Y63 priming site, phosphomimetic substitutions were made, where alanine mimics the unphosphorylated state and glutamic acid mimics the phosphorylated state [83]. The TP proteins of several species have amino acids near the priming site that have hydroxyl-containing side chains, and their phosphorylation state is unknown [16,17]. To determine if these sites in the human HBV TP domain are perhaps phosphorylated in vivo, four substitution mutants were made: T60A and T60E single-substitution mutants and S64A/S65A/T66A and S64E/S65E/T66E triple-substitution mutants. They showed interesting phenotypes, including stronger-than-wild-type RNA binding and RNA packaging [39]. One interesting possibility is that transient phosphorylation at these sites in vivo could facilitate RNA binding and RNA packaging. Both the glutamic acid substitutions (T60E and S64E/S65E/T66E) were defective in protein priming [39]. If they were phosphorylated in vitro, priming would be limited, and therefore, they are not likely phosphorylated in vivo during priming. When the probability of phosphorylation was predicted, several serine or threonine sites in the TP domain showed potential phosphorylation, but they were all downstream of the priming loop. HBV Pol is thought to be phosphorylated at two sites; these sites have not been identified [84].

W74 is needed for both the priming reaction and subsequent DNA synthesis, and is partially required for RNA binding [79,80]. Both Y63 and W74 are fully conserved among HBV isolates from both human and duck (Y96 and W107 in DHBV). The N42 site is 99% conserved in human HBV isolates and well conserved among mammalian HBV isolates. Despite its conservation, mutation to alanine (N42A) showed no effect [39]. Additionally tested, the amino acid insertion G44<KLLG>N45 in the priming loop disrupted DNA synthesis (not shown) [41]. This insertion may disrupt function through misfolding. Four internal deletions spanning different regions of the priming loop (Δ39–47, Δ48–55, Δ56–60, and Δ70–80) had no priming activity but could all bind and package RNA to some degree [39]. In general, the defects in the priming loop affect priming and downstream elongation, but not earlier steps.

Two pairs of mutations were tested in DHBV, H73A/K75A and K75/D87A, which are partially conserved with human HBV. Neither showed a defective phenotype [38]. Several candidate amino acids that would be profitable to test include those that are 100% conserved in this loop: G61, L62, N71, and P77 (Figure 3c). They likely have an important function that may be revealed by mutational analysis.

On a related note, even mutants that do not contain the Y63 priming residue can function in protein priming if supplied with manganese. The ability is likely due to cryptic priming (priming at an unspecified alternative site). Cryptic priming is known to occur when the standard priming site is removed, especially in the presence of manganese [43,85].

#### 3.3.3. Mutations in Helix 4

Helix 4 is the least well conserved subdomain among human HBV isolates (Figure 4a). It spans residues from 82 to 96 in human HBV Pol. Residues within this subdomain have been tested in several studies, but mutations in these subdomains have not revealed any associated defects in function [36,80,86]. This helix may be structurally important but not functionally important. Further testing will reveal if this is the case.

#### 3.3.4. Mutations in the L2 Packaging Loop

The L2 packaging loop is important in early stages of the replication cycle when the polymerase protein interacts with the viral RNA. It spans residues 97 to 134, and its main function is RNA packaging.

Most of the L2 loop substitution mutants could bind to the ε RNA (Figure 4a). The mutants with defective RNA binding phenotypes are the substitution mutant R114E and large internal deletion mutants Δ101–115 and Δ116–130 [39]. The R114E mutant reverses the charge from basic to acidic and affected all tested functions. Positively charged basic residues can interact with the negatively charged phosphate backbone of RNA, which may be the mechanism of RNA binding decrease for R114E. Considering the drastic change from arginine to glutamic acid, future experiments may reveal if an alanine substitution might also show a strong phenotype, as the defect may be due to the change in charge or some other chemical nature of the side chain. Two internal deletions that removed either half of the L2 packaging loop, Δ101–115 and Δ116–130, showed inactivity in all tested functions [39].

The L2 loop contains several mutants with defective RNA packaging phenotypes. Most of the mutations in this loop affect RNA packaging and subsequent steps [38,41,79,80,86,87]. R105 has been tested within double mutants (together with K104 or K108), but the defective phenotypes in these double mutants were attributed to R105 alone since it alone retained the defective phenotype when tested individually. The R105A mutant was defective in RNA packaging, priming, and DNA synthesis [38,79,86]. K130L blocked RNA packaging as well [87]. A double mutant Y133S/P134A blocked DNA synthesis, but the single mutant Y133A did not [80,86], indicating a potential role in DNA synthesis in these residues that has not been specified. Overall, the L2 packaging loop contains several amino acids important in RNA packaging.

Potential candidates for mutational analysis in the future include the following, all of which are 100% conserved: P112, A113, F115, P117, and P124 (Figure 3c). Mutational analysis may reveal important functions in this loop at these positions.

#### 3.3.5. Mutations in Helix 5

Helix 5 is the most conserved helix between duck and human HBV, especially near the T3 motif (Figure 4a, bottom right). It is predicted to contain one large helix that spans amino acid positions 135 to 152 [39]. Helix 5 contains a crucial tryptophan at position 147. Y147 is important for RNA binding, RNA packaging, protein priming, and DNA synthesis [37,78,79,80]. Mutating L148 as a double mutant with Y147 also showed defects; however, the 100% conserved residue L148 has not been tested as a single mutant (Figure 3c). H140A also showed partial defects, in RNA binding and protein priming [39].

Residues H140 and Y147 may influence priming by coordinating metal ion cofactors—Y147 affected magnesium priming only, and H140A contrastingly showed a strong defect in manganese priming [39,79,80]. These two mutants are located seven amino acids apart, together on a predicted helix. Interestingly, the side chains of both Y147 and H140 would face the same direction, considering the standard 3.6 amino acids per turn in an alpha helix. The normal in vivo priming reaction occurs using a magnesium cofactor, though in vitro can also use manganese as an alternative [44]. This reaction is technically a nucleotide transferase reaction because it occurs in a non-template-directed fashion. Thymidine is the preferred substrate for manganese priming, with which the enzyme can synthesize poly(T) strands of greater than two hundred bases in length. Nothing is known of the use of manganese priming in the in vivo replication cycle of HBV.

In summary, Y147 is significant in all HBV Pol functions, and Helix 5 may participate in metal ion coordination during protein priming, with the H140 and Y147 side chains affecting magnesium- and manganese-directed priming. One potentially important amino acid residue that is 100% conserved across all known hepatitis B viruses is L151. It has not yet been evaluated in a mutational analysis.

#### 3.3.6. Mutations in the RNA-Binding T3 Loop

The T3 loop is important in RNA binding, as revealed by extensive mutational studies [20,25,37,38,78,79,80,86,87]. It corresponds to positions 153 to 174 in the TP domain (positions 140 to 200 in DHBV Pol). Within the T3 loop is a short region known as the T3 motif, which has a highly conserved amino acid sequence between mammalian and avian HBV (Figure 4a). The T3 motif is thought to interact with a downstream sequence in the reverse transcriptase domain of the polymerase, RT1, to facilitate RNA binding [25,37]. The T3 loop is predicted to form beta sheets, which exist as pairs of at least two beta strands (Figure 3c and Figure 4a). The two prominent predicted beta strands may fold together forming an antiparallel beta sheet. The beta strand that encompasses the T3 motif may reversibly interact with the RT1 motif and the second beta strand in the T3 loop (Figure 4a). In this way, RNA binding may be controlled at the level of intramolecular interactions within the HBV Pol.

Several double mutants were tested in the T3 loop. K153A/A154G showed mild RNA binding defects, although the RNA packaging levels were unexpectedly normal [79]. G155/I156, when tested as a double mutant, showed either partial or complete inhibition of RNA binding, weak RNA packaging, and defective protein priming. However, when these residues were mutated individually at G155 or I155, no defects were observed [25,78,79,80]. Similarly, the double mutation of L157 and Y158 was defective in RNA binding and RNA packaging [78], but individual mutants at these locations were not defective [25,37,80].

The residues K159 and R160 in the T3 motif may be the most important residues in the entire TP domain—they are essential for all HBV Pol functions when mutated together. This has been observed in several studies using both HBV Pol and DHBV Pol [20,25,37,38,79,86,87]. One study, however, showed no defect in either RNA packaging or DNA synthesis with human HBV Pol when both residues were mutated to alanine, and no defect with single mutations K159E or R160A [86]. It was suggested that these differences in phenotype can be attributed to differences between mammalian and avian hepadnaviruses, though K159 and R160 are critical for both HBV and DHBV Pol function. Although several double mutants at K159/R160 were shown to be defective in RNA binding, only one single mutant showed any defects—R160 was defective in RNA packaging when mutated to isoleucine or glutamic acid (but not to alanine) [86,87]. When single mutants are created, perhaps there is little defect in RNA binding because only larger changes can disrupt the formation of beta sheets.

Other mutants downstream of the T3 motif were not tested or found to be essential for the earliest stage, RNA binding, but T162 has been shown to be necessary for RNA packaging [88], and Y173 is critical for RNA packaging, priming, and DNA synthesis [79,80]. Continued testing outside the T3 motif may reveal important information about this subdomain of the polymerase. For example, F168 and G170 in the T3 loop are 100% conserved among mammalian and avian HBV Pol isolates but have not been tested by mutational analysis (Figure 3c).

#### 3.3.7. Mutations in Helix 6

Helix 6 extends from position 175 to 192 in HBV Pol’s TP domain. It has not been mutated extensively. One mutant, W175A, showed no defect in function (Figure 4a). A four-base insertion D178<RSFD>L179 disrupted the DNA synthesis of the virus (not shown); however, this insertion may not be found in the course of a natural infection [41]. Due to the lack of defective phenotypes in this region, Helix 6 is thought to have no importance in the functionality of the virus but is perhaps essential for the structure of the polymerase, perhaps acting as a structural anchor for the spacer domain. The spacer domain is the least well-conserved domain of HBV Pol and perhaps the least important functionally; it is thought to provide flexibility for the RT domain, which remains attached throughout synthesis [18]. Helix 6 may provide the needed stability that allows a flexible spacer to bridge from the TP to the RT domain.

#### 3.3.8. Notes about Mutational Analyses

Figure 4a summarizes the mutations created in human HBV, and also includes mutants created in duck HBV where the amino acids are conserved. However, several mutations have been analyzed in DHBV where the tested amino acid is not conserved in human. For example, E124A/R128A, K137A/K141A, R173A/E176A, and K186A in DHBV are not conserved in human HBV, but they all show a defect in DNA synthesis [38]. Such mutations may tell a useful story about human HBV function if analyzed carefully.

It should be noted that mutations in the TP domain developed in mutational studies such as those reviewed here are usually generated in *trans*-complementation systems, where HBV Pol is not expressed from the same DNA as other open reading frames (ORF). On the native HBV DNA genome, all other ORFs overlap with HBV Pol. In the TP domain, HBV core antigen (HBcAg) overlaps with the first 14 amino acids [89], HBV e antigen (HBeAg) overlaps with the first 47 amino acids, and HBsAg overlaps 169C to the end (Figure 1 and Figure 4). Due to this overlap, mutations in the code for one protein may cause mutations in another. For example, mutations in the RT domain of HBV Pol (the target of antivirals) have been shown to affect HBsAg (the antigenic target of the HBV vaccine) and vice versa. This can cause functional changes in viral fitness, immune interactions, and drug resistance [90,91,92]. However, due to the lack of TP domain sequencing data, no information has been published about ORF overlap between the TP domain of HBV Pol and HBcAg/HBeAg.

This overlap denotes additional selection pressure against natural mutations in both ends of the TP domain. However, it is interesting to note that important loop subdomains in the TP domain are largely free of overlap with other ORFs, making these loops more genetically flexible. Of those mutations tested that would overlap with other ORFs (R9A/R10A, D15A/D16A, E22A/E23A, D30A/E31A, E31K, R35A/R36A, E39A/D40A, and N42A), all would cause changes to the HBeAg and/or the HBcAg if created in native infection systems [39,86]. However, this should not diminish the relevance of these laboratory-produced mutants in defining functional amino acids for the TP domain.

In addition to the TP domain mutants discussed here, there are, of course, important regions throughout HBV Pol whose mutation demonstrates the functional activities of HBV Pol (such as the YMDD active site located in the RT domain and the DEDD box found in the RNase H domain). Such mutations are also significant in HBV drug resistance [9] and are not discussed here. Importantly, mutations that lead to drug resistance may occur in the TP domain; however, this domain is not usually sequenced when testing for the mechanism behind observed resistance [10]. Therefore, if screening for drug resistance were to include the functionally important TP domain, additional insights would be gained about resistance mechanisms.

## 4. Conclusions

Although current drugs exist that target HBV Pol, they are not curative. Therefore, knowledge about the structure and function of the virus’ only catalytic protein, HBV Pol, will inform future treatment development. The structure of HBV Pol includes four domains: the TP, spacer, RT, and RNase H domains (Figure 1). Much information is already available about the structure and function of the RT and RNase H domains of HBV Pol due to its homology with the well-studied HIV. Indeed, all current HBV-specific drugs, except immuno-based drugs like interferon and thymalfasin, target the RT domain. However, the TP domain has not previously been studied to the same degree. The TP domain is known to be essential for HBV Pol function and has no structural homologs in humans, organisms, or other known viruses. Thus, the TP domain is an ideal target for novel therapeutics. A coordinated effort is needed to take candidate drugs from models, to in vitro tests, to cell culture or animal model systems, and on to clinical trials. It is hoped that a hypothesis-driven targeting of the HBV TP domain can begin, utilizing the mapping provided in this review.

The TP domain is essential for most known polymerase functions, which are (in order of occurrence): binding to the ε RNA, packaging of viral pgRNA into the nucleocapsid, priming the initial synthesis of DNA through the unique process of protein priming, and yjr continued DNA synthesis of both the minus and plus strands (Figure 2). Although the exact structure of the TP domain is still unknown, recent advances and standard assays have been able to map specific functions onto the structure. The TP domain contains seven subdomains, with three conserved loops that control RNA binding (T3 loop), RNA packaging (L2 loop), and protein priming (priming loop). The helical subdomains are less well conserved and less functionally important (Figure 3c and Figure 4a).

The findings summarized herein, which describe and map the functions of the TP domain of HBV Pol, will inform therapeutic drug development and basic HBV biology. Additional studies of the TP domain are essential to uncovering the overall functionality of the HBV Pol, its three-dimensional structure, and the ability to target critical regions such as individual TP domain loops.

## Figures and Tables

**Figure 1 viruses-12-00570-f001:**
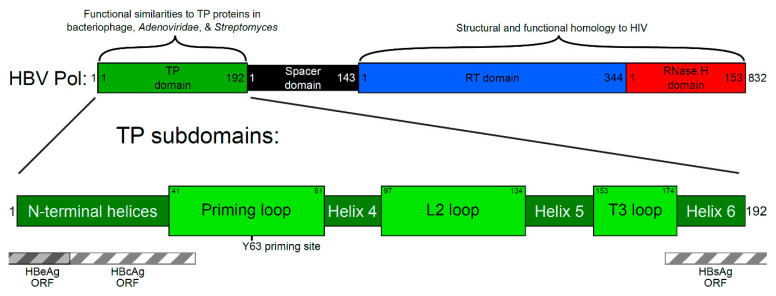
Schematic representation of hepatitis B virus polymerase domains and terminal protein (TP) subdomains. The polymerase protein is responsible for DNA synthesis, which is carried out by the catalytic reverse transcriptase (RT) domain. While the RNA template is copied into viral DNA, the RNase H domain degrades the RNA template. The TP domain acts as a primer for the initial DNA synthesis steps, and the DNA remains attached to the TP domain throughout synthesis. The spacer domain is thought to allow for flexibility while the TP is attached at one end of the nascent viral DNA and the RT domain synthesizes the other end. Regions of similarity with other proteins are highlighted. Within the TP domain (green), the function of the helical subdomains is likely to provide structure, while the loop subdomains are involved in activities critical to the viral replication cycle, such as protein priming that initiates from the tyrosine at position 63 (Y63). The regions of the TP domain that overlap with other open reading frames (ORF) are shown as striped boxes. These regions are under additional selection pressure when mutations occur. HBV: hepatitis B virus; HIV: human immunodeficiency virus; Pol: polymerase.

**Figure 2 viruses-12-00570-f002:**
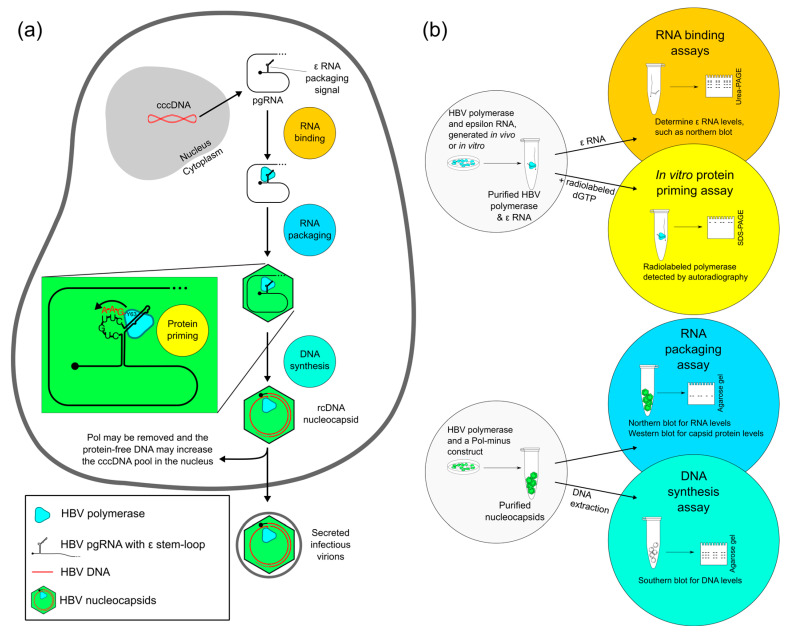
Polymerase-dependent functions of the replication cycle of hepatitis B virus (HBV) and four functional assays of HBV polymerase (HBV Pol) activity. (**a**) Starting with covalently closed circular DNA (cccDNA), which is located inside the nucleus of infected liver cells, the pre-genomic RNA (pgRNA) is transcribed by the host cell. Next, pgRNA is sent to the cytoplasm and translated into viral proteins. The translated HBV Pol binds to pgRNA at the 5′ epsilon stem-loop structure. The interaction between the HBV Pol and the pgRNA facilitates the encapsidation (packaging) of the HBV Pol and pgRNA. Priming most likely occurs after RNA packaging, within the nucleocapsid. Using a free hydroxyl group of the tyrosine at position 63 (Y63), HBV Pol performs the dual functions of primer and polymerase. The DNA strand remains attached to HBV Pol while synthesizing the initial GAA nucleotides. HBV Pol and these bases then change templates to the 3′ end of pgRNA and carry on minus-strand DNA synthesis. As the template pgRNA is copied, it is degraded by the RNase H activity of HBV Pol. This ssDNA-containing nucleocapsid will mature as the second strand of DNA is copied by HBV Pol, forming relaxed circular DNA (rcDNA). This rcDNA-containing nucleocapsid is the infective particle. (**b**) In RNA binding assays, the HBV Pol can be purified with in vitro synthesized RNA or epsilon RNA from cell culture (referred to as in vivo), which is co-purified with HBV Pol. Together, they can be evaluated for RNA binding activity by measuring RNA levels. Priming is measured from polymerase-epsilon constructs that synthesize DNA in the in vitro priming assay. Due to protein priming, HBV Pol becomes covalently labeled with the radionucleotides and can be detected by autoradiography. RNA packaging assays use HBV Pol and a polymerase-minus construct. The RNA in the purified nucleocapsid is compared to capsid protein levels, a ratio that describes RNA packaging levels. DNA synthesis can also be measured from purified nucleocapsids using HBV-specific Southern blotting. ε: epsilon RNA secondary structure motif.

**Figure 3 viruses-12-00570-f003:**
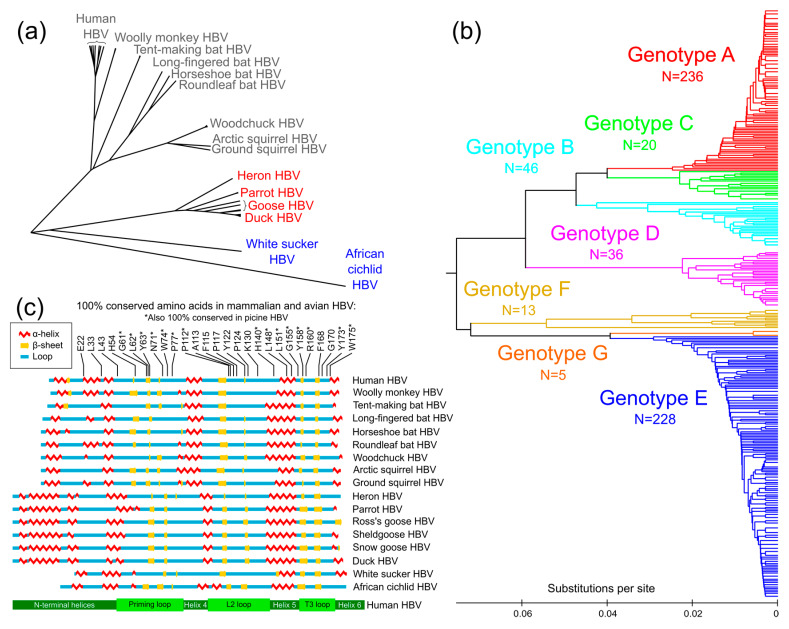
Evolutionary relationships and conservation among hepatitis B virus (HBV) isolates. (**a**) Amino acid sequences from 66 piscine (blue), avian (red), and mammalian (gray) HBV isolates were aligned and compared phylogenetically. Branch lengths represent the evolutionary distances between isolates of HBV that infect different host species. (**b**) Genotypes of human HBV isolates are grouped phylogenetically. Amino acid sequences from 584 human HBV isolates were compared, using reference sequences for genotypes A through G for determining genotype branching (colorized branches). Genetically identical isolates were consolidated into single lines. For both (**a**) and (**b**), the evolutionary history was inferred using the unweighted pair group method with arithmetic means. Trees are drawn to scale, with branch lengths in the same units as those of the evolutionary distances used to infer the phylogenetic tree. Evolutionary analyses were conducted in MEGA [55]. (**c**) Alignment of secondary structure predictions from representative HBV species of mammalian, avian, and piscine isolates. Alpha helices and beta sheets are shown for each sequence, and similar secondary-structure patterns were aligned based on three unstructured loops (regions between helices). The amino acid positions that are 100% conserved among mammalian, avian, and piscine HBV isolates are numbered and shown highlighted at the top, along the human HBV sequence for genotype D. A basic map of the subdomains of the human HBV TP domain is shown for reference, at the bottom in green.

**Figure 4 viruses-12-00570-f004:**
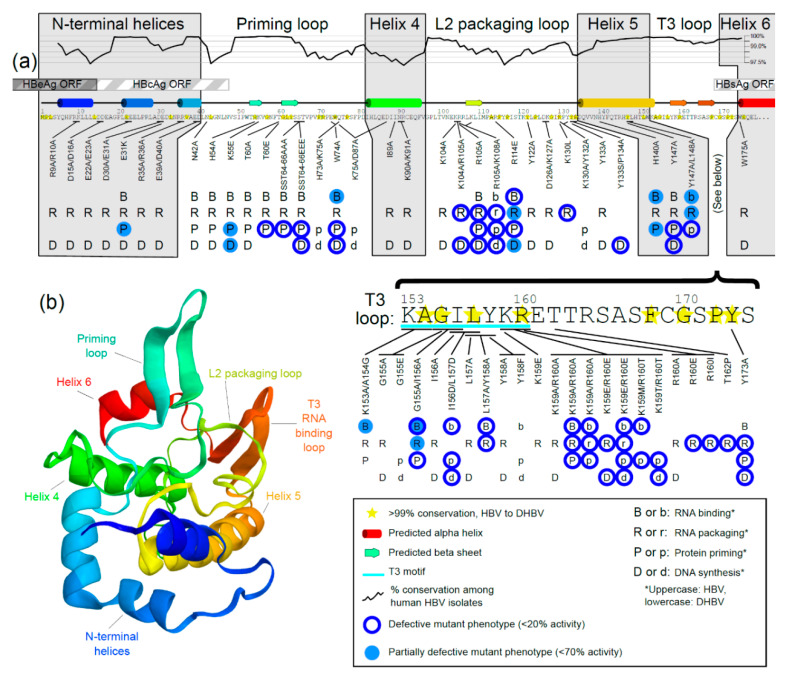
Two-dimensional and three-dimensional interactions of the terminal protein (TP) domain of the hepatitis B virus (HBV) polymerase protein. (**a**) Amino acid sequence for the TP domain, numbered according to genotype D. Overlap with other open reading frames (ORF) is indicated by striped bars. The TP domain is grouped into seven subdomains according to secondary structure predictions, with the helical regions in gray boxes. All known HBV (uppercase) and duck HBV (lowercase) mutants that have been tested are marked to represent the four tested functions. If a letter is absent, the mutant was not tested in that assay. Defective phenotypes are highlighted by blue circles representing either an extensive loss of function (<20% activity) or partial loss of function (<70% activity), whereas mutants exhibiting a wild-type phenotype are not circled. The T3 motif has been extensively tested in mutational studies. Yellow stars show TP domain residues with >99% homology to duck HBV (DHBV). Homology among human samples is graphed above the sequence (black line), where peaks indicate regions of high homology. Homology is shown as percent conservation, a moving average of amino acid homology among 584 isolates of human HBV in 11 bp windows (five amino acids upstream and downstream from each position). (**b**) The three-dimensional structure of TP as predicted using QUARK. The seven predicted subdomains of TP are indicated and color-coded. The functional loop subdomains extend outward.

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
