# Peer review of "Molecular, Evolutionary, and Structural Analysis of the Terminal Protein Domain of Hepatitis B Virus Polymerase, a Potential Drug Target"

_viruses, 2020, doi:10.3390/v12050570_

Round 1
Reviewer 1 Report
In this manuscript, Buhlig and colleagues aimed to review the current knowledge about the biological functions of the Terminal Protein Domain of the Hepatitis B Virus Polymerase. The most important and frequently cited original reports about the HBV TP domain are comprehensively covered and well described by the authors. Also, the figures are complementing the explanations in the text sufficiently. Since there is no PubMed listed review article available that solely focussing on the in-depth knowledge of the HBV TP domain, this review article will be beneficial for the community of HBV researchers.
Collectively, the manuscript is of interest and written well; however, there are minor concerns that are recommended to be addressed by the authors.
- Thirty-seven out of 44 cited references are older than five years and 25 older than ten years. It is impossible to cover all publications related to the topic. However, it would complete the review article if the latest trends and current developments in the field of HBV TP domain research would be emphasized more. Related to this point, in manuscript line 165, the authors state that “…fastidious requirements hampered in vitro studies of human HBV Pol priming until recently, when a system reproduced human HBV polymerase activity in epsilon-dependent protein priming [20].” The citation #20 is from the year 2012. The authors may reconsider their assumption that an eight years old publication is a recent development. Please rephrase and double-check for further, more recent publications to be addressed by the review.
- In general, more references should be cited. 50-100 references are appropriate for a review article. A review article is supposed to be an equal platform to all ideas controversially discussed and sensitive to all existing literature promoting competing hypotheses. Fourteen out of 44 cited references are from the manuscripts corresponding author (Clark D.) and a closely related lab of Hu J. Since Clark and Hu are both leading experts in the small community of HBV TP domain researchers, a high ratio of self-promoting citations is understandable. However, the authors may reconsider if further critical publications are available and should be discussed more controversially.
- Across the manuscript, the authors are convincingly emphasizing very frequently the high potential of the HBV polymerase TP domain as a target for drug development. However, strategies to inhibit the TP domain functions are not discussed. It would be informative for the reader to have a brief overview of the current state of inhibitor development, specifically targeting the TP domain. Regarding this, shortcomings and future perspectives of TP-domain inhibitor development should be included covering, for example, the following questions: Are there any chemicals known to target and inhibit TP-function? What is needed to translate the comprehensive knowledge about the TP domain functions into the development of TP domain targeting inhibitors? The mentioned assays like RNA binding, protein priming, RNA packaging assays, etc. as well as in silico modeling and docking studies are all hypotheses driven, cell-free, and target-based and rather allowing for low-throughput drug screenings. Is it possible to upscale those target-based assays to medium- or high-throughput drug screening approaches for first-in-class drug identification? Compared to cell-free assays, cell-based assays provide the physiopathological biology of cellular host-factors that interact with the HBV pgRNA – polymerase complex as well as giving an idea about the availability of a drug to target the HBV pgRNA – polymerase complex within the cell/nucleocapsid. Therefore, more empirical approaches of cell-based assays are known to be more efficient in identifying first-in-class drugs. Are there any cell-based assays described in the literature, that would be instrumental in screening and identifying drugs interfering in the function of the TP-domain promoting pgRNA-Pol interaction (pgRNA-Pol binding, pgRNA encapsidation, DNA synthesis)?
- Figure 3:
- The legend is lacking information to fully understand what is shown in the Figure. Please be more precise
- In text line #245-247, the authors mention that “…the TP domain alone allowed for distinct branching according to these classical genotypes (Figure 3A), demonstrating that the TP domain diverges along with genotype groups, a sign of its evolutionary importance.” It is unclear/confusing, where in the Figure the distinct branching of the TP domain is shown. Please clarify.
- Where in the text is Figure 3B explained? Please clarify.
- Original articles first describing the sequences that were used for the analysis of the evolutionary relationships and conservation among hepatitis B virus (HBV) isolates should be cited.
Reviewer 2 Report
The target of current HBV drug treatment is limited to the reverse transcriptase domain. The authors focused on the terminal protein (TP) as the next target for drug development. The authors reported overview of HBV polymerase, and detailed summary of current knowledge of TP. The structure, and function of TP was explained. Then, mutations of amino acid in all the parts of TP were shown, and dysfunction of replication cycle caused by these mutations were explained in details.
Major point:
In page 7, the authors mentioned that divergence of TP domain is a sign of evolutionary importance. On the other hand, the authors mentioned that highest conservation in the loops may be more evolutionarily important. These sentences seem contradictory.
Reviewer 3 Report
HBV polymerase, the largest viral protein and the sole known viral enzyme, plays multiple roles in viral replication, including pgRNA binding and packaging, protein priming and DNA synthesis. Distinct from the RT and RNase H domains that have been targeted by approved or trialed drugs, the TP domain shows considerable function however not targeted yet. Therefore, more studies could focus on this unique subdomain, despite the lack of structural information.
In this review, Buhlig et al. summarized all functional assays of viral polymerase, phylogenetic analysis among the TP domains in HBV isolates and mapping in all key residues of the TP protein. This review was finely organized, well written and provided comprehensive understandings for enlightening readers to facilitate future science. However, some points have to be argued to make the paper superior.
Major comments:
- The authors should implement that the TP domain overlaps two open reading frames (ORF) of HBV e antigen and preS1 protein and conclude that all related mutagenesis in the TP domain showed effects on actual TP function without affecting HBe and HBs by tracking those mutations (R9/R10, D15/D16, E22/E23, D30/D31, E31, R35/36, E39/D40, N42) in the original literature. Although it is probable when mutations were experimentally introduced in most studies, amino acids in HBe and HBs proteins were not changed by choosing the correct nucleotide substitutions, the same mutations that occur naturally during viral replication may also alter HBe and HBs. I am glad to see that the authors have mentioned this point from line 499 to line 504, however, I expect more citations and the above confirmation.
To emphasize this point, I suggest that the authors could draw a fraction of the HBe ORF (overlapping with 1-47 residues of the TP domain) and HBs preS1 ORF (overlapping with 181~192 residues of the TP domain) in Figure 1 and Figure 4a, as I notice that the HBe shares the entire region of the N-terminal helices but almost does not cover the priming loop, similarly the preS1 coves the helix 6 but has no relation with the T3 loop. This phenomenon seems to make important domains, the priming loop and the T3 loop, much more flexible. Can you comment on this point in the review?
- I doubt whether and why three-dimensional interactions of the TP domain (Figure 4b and 4c) is necessary for this review. Firstly, unlike Figure 4A which is based on the assays, Figures 4b and 4c showed a predicted 3D structure using QUARK and a potential intramolecular interaction using RaptorX, however, these in silico results have not been tested yet and thereby not suitable for presentation in a review paper. In addition, the TP domain is not a single protein, it seems not the best way to model it without adding the rest three domains within the polymerase. Second, the authors only mentioned Figure 4b in line 284-285 and Figure 4c in lines 287 and 347 but Figure 4a occupied the whole chapter 3.3 lasting from line 316-512. This makes 4b and 4c thin and weak. Lastly, Figure 4a dominates the figure, all of which provided numerous information for readers. If it is a research article, Figures 4b and 4c are suggested to be either a single or a supplemental figure. If the authors could not provide more information, 4b and 4c are suggested to be deleted in this review.
Minor comments:
- In the title, the word “interaction” reads too abstract. One option is to rephrase it such as “Assays and functional mapping” etc.
- The approved interferon-a and pegylated interferon-a have multiple targets (immunomodulator, etc.) and thereby not polymerase-specific. The authors should present “except interferon-a” in many paragraphs and avoid presenting “all current drugs are directly against polymerase”: in lines 12-13; lines 31-32; line 514; line 519. Please correct it.
- In response to the lines 90-93, how about cryo-EM or NMR techniques in dissecting polymerase structure? Any other methods are available instead of classical crystallography?
- Also, cite Figure 2a somewhere in the introduction session.
- In Figure 2a it is difficult to read which strand is pgRNA, minus-stranded DNA or plus-stranded DNA. It is better to use two colors for RNA and DNA, respectively.
- In line 111, the authors drafted “the replication cycle of hepatitis B virus (HBV)…” however drew a part of the life cycle”. It seems better to say “polymerase biology in HBV replication…” or something else.
- In lines 204-206, the removal of HBV polymerase before cccDNA formation is not shown in Figure 2a.
- Here is a discrepancy between Figure 1 with Figure 4a, in terms of the length of the TP domain. 192 amino acids were shown in Figure 1, but 180 (anyway less than 190) as shown in Figure 4a and it is difficult to read so small characters.
- Regarding circled defective phenotypes, the authors mentioned those circles represent impairment or loss of function depending on the mutant. Thus for better reading, the authors should use two colorized circles: blue ones for defective mutants, and red for dead mutants according to the experimental data (northern, Southern, western blot) in the original publications. It is important to show defective and dead mutants that impact viral replication to different extents.
Dear Prof. Dr. Clark,
It is nice to read your manuscript. Please attached find my comments.
Stay healthy and happy during this special season.
With my best regards,
Reviewer
Round 2
Reviewer 2 Report
The manuscript has been improved.